



# Radiative closure tests of collocated hyperspectral microwave and infrared radiometers

Lei Liu[1], Natalia Bliankinshtein[2], Yi Huang[1], John R. Gyakum[1], Philip M. Gabriel[3], Shiqi Xu[2], Mengistu Wolde[2]

[1]Department of Atmospheric and Oceanic Sciences, McGill University, Montreal, Quebec, Canada
[2]Flight Research Laboratory, National Research Council Canada, Ottawa, Ontario, Canada
[3]Horizon Science and Technology, Wolfville, Nova Scotia, Canada

*Correspondence to*: Lei Liu (lei.liu5@mail.mcgill.ca)

**Abstract.** Temperature and water vapor profiles are essential to climate change studies and weather forecasting. Hyperspectral
instruments are of great value for retrieving temperature and water vapor profiles, enabling accurate monitoring of their changes. Successful retrievals of temperature and water vapor profiles require hyperspectral radiometer measurement accuracy. In this study, the radiometric accuracy of an airborne hyperspectral microwave radiometer, High Spectral Resolution Airborne Microwave Sounder (HiSRAMS), and a ground-based hyperspectral infrared radiometer, Atmospheric Emitted Radiance Interferometer (AERI), is simultaneously assessed by performing radiative closure tests under clear-sky conditions.
As an airborne instrument, HiSRAMS has two radiometers measuring radiance in the oxygen band (49.6-58.3 GHz) and water vapor band (175.9-184.6 GHz) for zenith-pointing and nadir-pointing observations. AERI provides ground-based, zenith-pointing radiance measurements between 520 and 1800 cm$^{-1}$. A systematic warm radiance bias is present in the temperature-sensitive channels in AERI observations in the window band. Upon removal of this bias, improved radiative closure was attained in the window band. The brightness temperature (BT) bias in nadir-pointing HiSRAMS observations is smaller than
at the zenith. A novel but straightforward method is developed to diagnose the radiometric accuracy of the two instruments in comparison based on the relationship between radiometric bias and optical depth. Compared to AERI, HiSRAMS demonstrates similar radiometric accuracy for nadir-pointing measurements but exhibits relatively poor accuracy for zenith-pointing measurements, which requires further characterization. Future work on temperature and water vapor concentration retrievals using HiSRAMS and AERI is warranted.

## 1 Introduction

Accurate long-term measurements of the vertical distributions of temperature and water vapor are crucial for climate change analysis, climate model validation, and weather forecasting. Radiosondes provide accurate, in situ temperature and water vapor profiles at high vertical resolution but are limited in spatial and temporal coverages. Remote sensing techniques have been developed to fill such data gaps (Aires et al., 2015; Blackwell et al., 2010; Delamere et al., 2010; Turner and Blumberg, 2018;
Warwick et al., 2022; King et al., 1992; Han and Westwater, 1995; Westwater, 1997; Turner et al., 2000). Hyperspectral



measurements, in which the vertical information of temperature and water vapor can be retrieved from different spectral channels (Smith et al., 2021), are valuable for sounding their vertical distributions (e.g., Divakarla et al., 2006; Turner and Blumberg, 2018). Spectral resolution (number of channels within a certain spectral range) is pivotal in determining the information content in such retrievals (Rodgers, 2000).

Both hyperspectral infrared and microwave radiometers can be employed to retrieve temperature and water vapor concentration profiles. A distinct advantage of microwave radiometers in retrieving temperature and water vapor profiles is their ability to sound through clouds, allowing for all-sky retrievals. However, the existing microwave radiometers typically have no more than 100 spectral channels (Blackwell et al., 2010; Hilliard et al., 2013), which is an order of magnitude less than infrared hyperspectrometers (Aumann and Strow, 2001; Carminati et al., 2019; Knuteson et al., 2004b). Thanks to the

advancement of digital polyphase Fast Fourier Transform (FFT) filter banks, hyperspectral microwave radiometers can now acquire a comparable number of spectral channels, which allows us to access and compare their temperature and water vapor profiling abilities as well as develop synergies between hyperspectral microwave and infrared radiometers. The High Spectral Resolution Airborne Microwave Sounder (HiSRAMS) is such a hyperspectral microwave radiometer, developed by Omnisys Instruments AB, National Research Council of Canada (NRC), and McGill University, under the sponsorship of the European

Space Agency (Auriacombe et al., 2022; Bliankinshtein et al., 2023b). As a prototype for possible future satellite missions, HiSRAMS' accuracy needs thorough assessment.

In this study, we focus on two hyperspectral radiometers: 1) HiSRAMS, operating in the microwave spectral range (49.6-58.3 GHz and 175.9-184.6 GHz for single-polarized observations), and 2) the Atmospheric Emitted Radiance Interferometer (AERI) operating in the infrared spectral range (520-3200 cm$^{-1}$). AERI is a well-tested instrument with good radiometric

accuracy (Knuteson et al., 2004a), which provides a benchmark comparison for the radiometric accuracy of HiSRAMS.

HiSRAMS, a payload mounted on a wing of NRC's Convair-580 research aircraft (Bliankinshtein et al., 2022), provides zenith-pointing (looking up) and nadir-pointing (looking down) observations or can be deployed on the ground for zenith-pointing observations. AERI is perpetually deployed on the ground for zenith-pointing observations (Knuteson et al., 2004b, 2004a). Both instruments have high spectral resolutions and mainly target the retrieval of temperature and water vapor profiles

with the potential of retrieving other trace gases. When airborne, HiSRAMS can take measurements at different altitudes. Such multi-altitude measurements yield more constrains of the detailed and extensive temperature and water vapor retrievals. In comparison, AERI has been demonstrated to be capable of retrieving temperature and water vapor profiles at high vertical resolutions, especially in the boundary layer (Turner and Löhnert, 2014; Turner and Blumberg, 2018).

The radiometric accuracy of the hyperspectral measurements is vital for successful retrievals. For example, in the optimal

estimation method (Rodgers, 2000), the ability of a hyperspectrometer to resolve the vertical distributions of temperature and water vapor can be measured by the Degree of Freedom for Signals (DFS), which is dependent on the characterizations of errors in both the hyperspectral measurements and the meteorological variables. Radiative closure tests can help determine the bias in the radiometer measurements and provide clues to their origins (Barrientos-Velasco et al., 2022; Clough et al., 1994; Delamere et al., 2010; Turner, 2003). In this study, we focus on clear-sky radiative closure tests to avoid uncertainties due to





the poor representation of clouds. Two primary objectives of this work include 1) the collection of collocated AERI and HiSRAMS radiance measurements under clear-sky conditions and 2) performing radiative closure tests for both radiometers and compare their radiometric accuracy.

## 2 Data and method

### 2.1 Datasets

Three clear-sky field campaigns (FC2021, FC2022, and FC2023) were carried out to collect hyperspectral measurements and to perform radiative closure tests of an AERI stationed on the ground and the HiSRAMS mounted on the NRC Convair-580 research aircraft (details listed in Table 1).

**Table 1. Summary of the three field campaigns.**

| Field Campaign | Date | Radiosonde | HiSRAMS | AERI |
|---|---|---|---|---|
| **FC2021** | 29 October 2021 | 14:21:57 - 15:59:32 UTC PWV: 0.69 cm | Ground-based measurements, pre-refurbishment, dual- and single-polarized (14:22:00 – 15:59:00 UTC) | Continuous ground-based measurements, every ~20 seconds |
| **FC2022** | 9 December 2022 | 18:57:33 - 20:08:47 UTC PWV: 0.37 cm | Ground-based measurements, after-refurbishment, dual- and single-polarized (18:45:37 – 20:10:34 UTC) | |
| **FC2023** | 11 February 2023 | 14:22:53-15:26:22 UTC PWV: 0.32 cm | Flight measurements at different altitudes, ground-based measurements before taking off (13:45:45 - 13:46:28 UTC) and after landing (16:35:24 UTC), single-polarized | |

Radiosonde measurements were collected (one for each campaign), together with the HiSRAMS (Figure 1a, 1b) and AERI measurements (Figure 1c). Ground-based zenith-pointing HiSRAMS measurements were archived in all three field campaigns. In the first two field campaigns, HiSRAMS collected longer ground-based records. In the final field campaign, HiSRAMS was mounted on the NRC Convair-580 research aircraft to gather ground-based zenith-pointing measurements before take-off and after landing, including airborne measurements at different flight altitudes. In all three field campaigns, AERI provided

continuous ground-based zenith-pointing measurements.



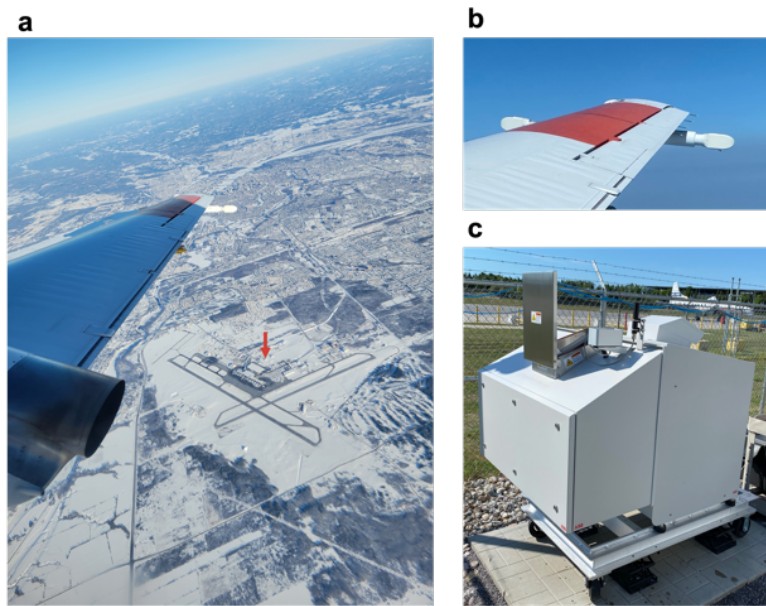

**Figure 1. (a, b) HiSRAMS mounted on the wingtip of NRC Convair-580 research aircraft for zenith-pointing and nadir-pointing measurements during the flights. The arrow in panel a indicates the location of AERI. (c) AERI on the ground with the hatch open, taking zenith-pointing measurements.**

### 2.1.1 Radiosonde temperature and water vapor profiles

The radiosonde used in this study was an iMet-4 from InterMet with 5% relative humidity uncertainty, 0.5 K temperature uncertainty below 100 hPa, and 1 K temperature uncertainty above 100 hPa. All the specified statistical uncertainties were at the 95% confidence level (see https://www.intermetsystems.com/products/imet-4-radiosonde/). These data were transformed into 3-sigma statistics for the radiative closure uncertainty analysis. The temperature and water vapor profiles from in situ radiosonde observations are considered to represent "truth" of atmospheric thermodynamic states (Figure 2); they are inputs to the radiative transfer models for testing the radiative closure. Table 1 lists precipitable water vapor (PWV) converted from radiosonde water vapor measurements in each field campaign. The small fluctuations in the temperature and water vapor vertical profiles, e.g. the temperature oscillating around 2.5 km for FC2022, have little effect on the radiance for AERI and HiSRAMS (not shown).

In the boundary layer, temperature inversions were present in all three field campaigns (see inset in Figure 2a), with two temperature inversions around 0.4 km and 1.2 km in FC2021, one temperature inversion around 0.5 km in FC2022, and one temperature inversion around 0.8 km in FC2023. Drier layers associated with the temperature inversions were also observed in all three field campaigns (Figure 2b). Based on the temperature, dew point temperature, and water vapor profiles, the cause of the temperature inversions was subsidence. The sources and features (such as the fine vertical structure) of the temperature and water vapor anomalies exhibited in these profiles are beyond the scope of this paper but warrant future analyses.





Hourly-mean atmospheric state profiles from the fifth generation European Centre for Medium-Range Weather Forecasts atmospheric reanalysis dataset, ERA5 (Hersbach et al., 2020), at nine grid boxes surrounding the field campaign location (latitude: 45.32°, longitude: -75.66°) were also included for analysis of the spatial variability of temperature and water vapor concentrations. Generally, the ERA5 hourly profiles agree well with radiosonde measurements, except that they do not resolve

the aforementioned dry layers, likely due to their limited vertical resolution. Considering this, we mainly use radiosonde-observed temperature and water vapor profiles for the radiative closure analyses.

A higher vertical resolution is employed in the boundary layer than that in the upper troposphere and stratosphere because AERI ground measurements are most sensitive to the lowermost layers. To avoid interpolating radiosonde measurements, the original temperature and relative humidity profile are updated every 5 seconds until the balloon reaches 3 km, then every 15

seconds until it reaches 10 km, and finally every 60 seconds until it reaches 20 km. The atmospheric conditions above 50 hPa (inclusive) from ERA5 are added to the top of the radiosonde measurements to form a hybrid full profile. Temperature and water vapor concentration at over 200 levels are inputs to the radiative transfer models.

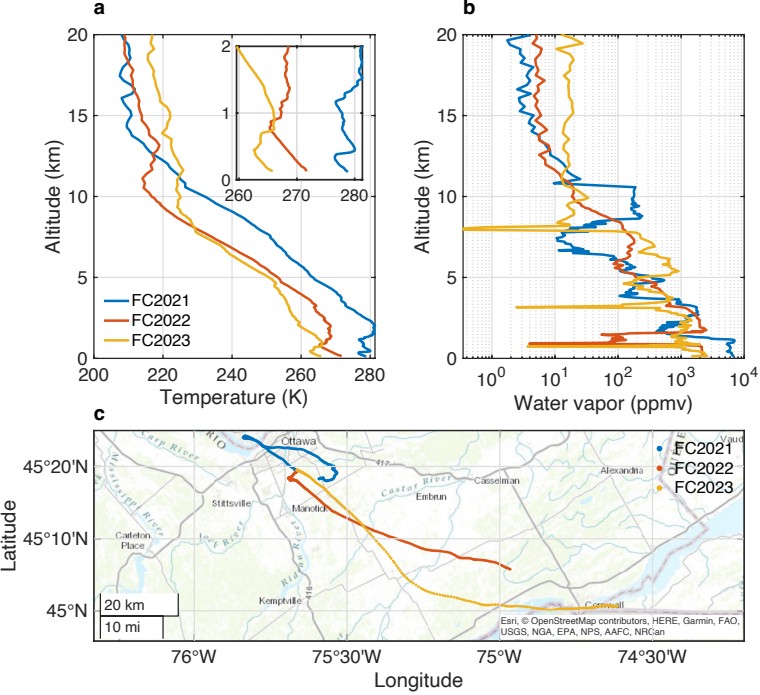

**Figure 2. Radiosonde in situ measurements of (a) temperature and (b) water vapor concentration profiles in the three field**
**campaigns, together with (c) radiosonde trajectories.**

## 2.1.2 AERI spectra

AERI measures downwelling longwave radiance (DLR) emitted from the atmosphere from 520 to 3200 cm$^{-1}$, with a field-of-view (FOV) of 2.6 degrees, a spectral resolution of 0.5 cm$^{-1}$, and a temporal resolution of 20 seconds (Knuteson et al., 2004b,





2004a). In each 20-second observation cycle, aside from taking sky-view measurements, AERI also calibrates against two
blackbodies, an ambient blackbody at the temperature of the surrounding air and a hot blackbody at a fixed temperature of 60
°C, to ensure the accuracy of the measured DLR. In this study, the focus is on the AERI Channel 1 observations from 520 to
1800 cm$^{-1}$.

Given AERI is most sensitive to atmospheric conditions in the boundary layer (Turner and Blumberg, 2018), accurate
representation of the near-surface temperature and water vapor concentration profiles is essential to the analysis of the
radiometric accuracy of AERI. A balloon launch exceeds one hour, during which the thermodynamic conditions may change
considerably. As a result, the original AERI-observed spectra with ~20 s sampling frequency are averaged over the period
from 2 minutes before to 8 minutes after the balloon launch to provide temporal sampling consistency between AERI
observations (shown in Figure 3) and radiosonde profiles.

The radiance in the $CO_2$ absorption band centered at 667 cm$^{-1}$ and the water vapor absorption band between 1400 and 1800
cm$^{-1}$ indicates the radiating temperatures of the near-surface atmosphere. The radiance differences shown in Figure 3
correspond to the different air temperatures during the three field campaigns. The generally low radiance in the window band
(800-1200 cm$^{-1}$) confirms a clear-sky condition during the three field campaigns. The radiance differences here indicate
different PWV values. The radiance differences in the water vapor absorption band between 520 and 600 cm$^{-1}$ also indicate
the different PWV: the low PWV value of 0.32 cm in FC2023 led to very low radiance values in this spectrum.

In summary, the differences between the AERI spectra from the three field campaigns are qualitatively consistent with the
differences in air temperature and water vapor concentrations.

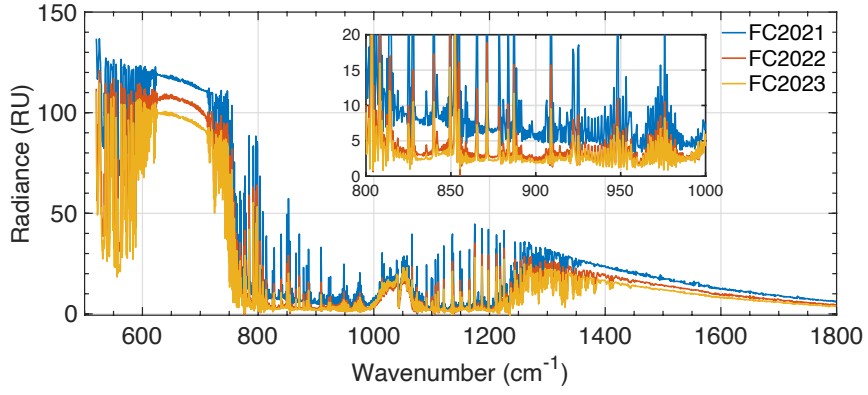

**Figure 3. AERI-observed spectra. The spectra are averaged over a period from 2 minutes before to 8 minutes after the time of the balloon launch.**

**2.1.3 HiSRAMS spectra**

HiSRAMS consists of two radiometers, one targeting an oxygen absorption band and the other a water vapor absorption band.
HiSRAMS can measure either single-polarized radiance over 49.6-58.3 GHz in the oxygen band and 175.9-184.6 GHz in the
water vapor band or dual-polarized radiance over 52.4-57.2 GHz in the oxygen absorption band and 178.8-183.5 GHz in the





water vapor band. Although dual-polarized measurements are valuable for characterizing radiance over water surfaces, this
study focuses on single-polarized observations because the nadir-pointing measurements from FC2023 were mostly over land.
With its Fast Fourier Transform (FFT) filter banks, the spectral resolution of HiSRAMS can be as high as 305 kHz
(Auriacombe et al., 2022). Noise in the brightness temperature (BT) measurements was reduced by averaging the
measurements to 6.1 MHz resolution, i.e., the radiance was resampled every 20 original HiSRAMS channels. Each HiSRAMS
radiometer has two FFT spectrometers: FFT0 and FFT1. For single-polarization observations, both FFT spectrometers have a
narrow overlapping frequency range. For dual-polarization observations, the two FFT spectrometers have identical spectral
ranges. HiSRAMS-observed spectra are calibrated regularly using measurements of a hot calibration load maintained at 80 °C
as well as an ambient calibration load.

Ground-based zenith-pointing HiSRAMS observations of single-polarized spectra are averaged over the entire observation
period, shown in Figure 4. As with AERI measurements, differences between HiSRAMS spectra in the oxygen and water
vapor absorption bands reflect the temperature and water vapor variations in the three clear-sky field campaigns. In the opaque
frequency range of about 56 GHz in the oxygen band, the effective emitting layer lies close to the surface, resulting in the
observed BT representing the near-surface temperature. Greater water vapor concentration results in a higher BT in the water
vapor band.

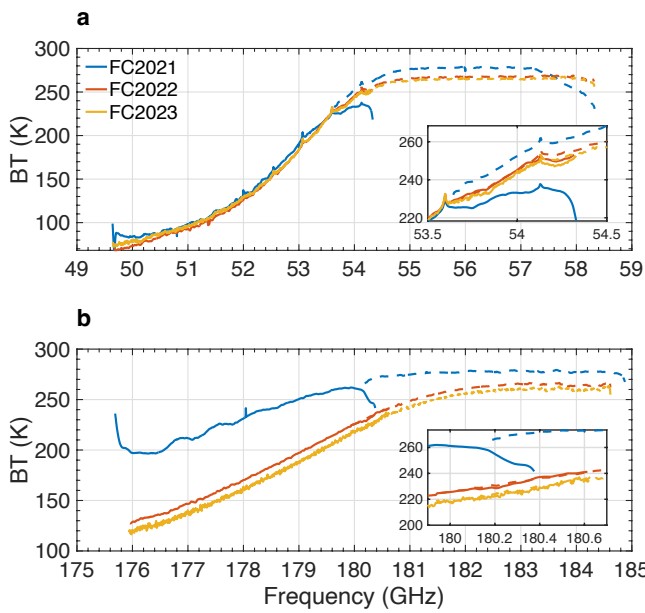

**Figure 4. HiSRAMS-observed ground-based zenith-pointing spectra in (a) oxygen band and (b) water vapor band. Solid and dashed
lines show the observed spectra from the two overlapping spectrometers, FFT0 and FFT1, respectively.**

In Figure 4, the observed spectra from the two FFT spectrometers are shown in solid lines (FFT0) and dashed lines (FFT1),
respectively. In FC2021, unphysical signals at the edge of the spectral range were detected, herein referred to as a "roll-off"
issue. This issue occurred in both FFT spectrometers, showing an overestimation of the radiance at the lower end of the



frequency range and an underestimation at the higher end. Hence, discrepancies between the two spectrometers were identified,

within the overlapping frequency ranges in the oxygen and water vapor absorption bands (see blue lines in insets in Figure 4).

One cause of the "roll-off" issue was attributed to incomplete image rejection in channels symmetric about the local oscillator

frequency (Xu et al., 2023). After a refurbishment in the summer of 2022 to improve HiSRAMS' image rejection behaviour

and to better characterize image response, the discrepancies between the two FFT spectrometers were significantly reduced.

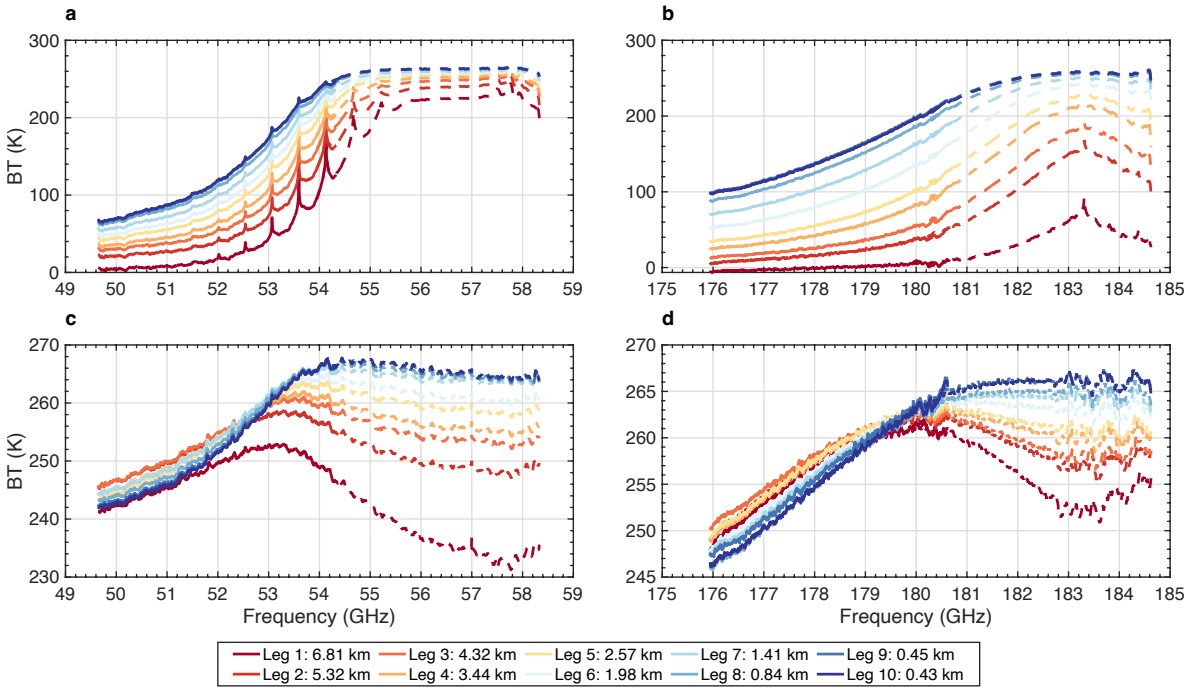


**Figure 5. HiSRAMS-observed spectra during FC2023 flights at different altitudes. Solid lines are for FFT0 measurements and dashed lines are for FFT1 measurements. (a, b) Zenith-pointing and (c, d) Nadir-pointing spectra in the oxygen and the water vapor band, respectively.**

The HiSRAMS flight measurements taken during FC2023 are shown in Figure 5. Zenith-pointing and nadir-pointing's

observations were taken over ten straight-and-level flight legs on February 11, 2023, with altitudes ranging from 429 m to 6.8

km. After the HiSRAMS refurbishment, the observed spectra in the overlapping frequency range have agreed well between

the two FFT spectrometers, in both the oxygen and the water vapor absorption band, at all flight altitudes.

In zenith-pointing spectra, the BT decreases with observation altitude in both oxygen and water vapor bands (Figure 5a, 5b)

because of the corresponding overall decrease in temperature (and water vapor), resulting in lower emitting temperatures with

increasing altitudes. In contrast, nadir-pointing spectra, in the strong absorption frequency range, e.g. 54-58 GHz in the oxygen

band and 181-184 GHz in the water vapor band, the BT decreases with altitude because the emitting layer goes higher

according to the $\tau = 1$ law, i.e. the altitude corresponding to $\tau = 1$ is where the weighting function peaks (Huang and Bani

Shahabadi, 2014), resulting in a lower emitting temperature, while in the weak absorption frequency range, e.g. 49.5-52 GHz





in the oxygen band and 176-179 GHz in the water vapor band, the BT increases overall with altitude, as a result of competing
contributions from the surface and from atmospheric emissions (Figure 5c, 5d).

## 2.2 Forward model

In radiative closure tests, the radiometric accuracy of a radiometer is verified by comparing its measurements to synthetic
spectra simulated by a radiative transfer model. The input of the temperature and water vapor concentration profiles to the
radiative transfer model is taken from radiosonde measurements, as described above.

### 190 2.2.1 AERI forward model

We use Line-by-Line Radiative Transfer Model Version 12.9 (LBLRTM v12.9, Clough et al., 2005) as the forward model for
AERI synthetic spectra simulation. LBLRTM-computed monochromatic radiance spectra were convolved with the AERI scan
function, enabling comparisons with AERI-measured spectra. Carbon dioxide concentrations (413.84, 418.75, and 419.72
ppmv), sourced from the global and monthly averaged marine surface values of the Global Monitoring Laboratory of the
National Oceanic and Atmospheric Administration (Lan et al, version 2023-06), remain constant across all vertical levels.
Ozone and methane concentration profiles were taken from the ERA5 reanalysis dataset and the Copernicus Atmosphere
Monitoring Service (CAMS) global atmospheric composition forecasts dataset (Inness et al., 2019), respectively. No CFC11
and CFC12 were prescribed in the synthetic spectra calculations.

### 2.2.2 HiSRAMS forward model

The HiSRAMS forward model (Bliankinshtein et al., 2019) consists of two major components, the Rosenkranz gas absorption
parameterization (Rosenkranz, 2017) and an efficient plane-parallel radiative solver that excludes multiple scattering, but
accounts for surface polarization. A sea surface emissivity model is used as an example boundary condition for nadir-pointing
measurements. The forward model was validated against the Monochromatic Radiative Transfer Model, MonoRTM (Clough
et al., 2005) and the Atmospheric Radiative Transfer Simulator, ARTS (Eriksson et al., 2011). To avoid uncertainty in regard
to the surface contribution in the closure tests, nadir-pointing measurement taken at the lowest flight altitude (429 m) were
employed as the boundary condition (i.e., elevating the surface to this altitude). The boundary emissions propagating upwards,
along with emissions from the atmosphere, constitutes simulate measurements at higher flight legs.

## 2.3 Radiative closure diagnosis

In this study, the radiance/BT bias is defined as the instrument-measured radiance/BT minus the forward model-simulated
radiance/BT, which provides a metric for evaluating the radiance closure (Eq. 1). The bias derives from the instrument
measurements and model simulations (Eq. 2). The instrument measurement uncertainty for AERI is 1% of ambient blackbody
radiance (3-sigma), which is its absolute radiometric calibration accuracy (Knuteson et al., 2004b). For HiSRAMS
measurements, if multiple individual measurements are averaged, the standard deviation of any individual measurements



during the whole observational period is considered to be the uncertainty of the HiSRAMS averaged measurements, which is
applied to HiSRAMS ground measurements in FC2021 and FC2022, and flight measurements in FC2023. If only the individual
observed spectrum is available, i.e. FC2023 HiSRAMS ground measurements, its uncertainty is determined by taking into
account the radiometric noise characterized by the noise-equivalent differential temperature, calibration load imperfections,
detector nonlinearity error, and instrument drift (Bliankinshtein et al., 2023a). Both the forward model uncertainty and the
uncertainties associated with the input variables contribute to the total uncertainty in model simulations. Input uncertainties
include radiosonde (instrumental) measurement errors and errors arising from the spatial variability of the inputs due to
radiosonde drift. We used the ERA5 hourly-mean profile in the rectangular region, including the balloon trajectory (Figure
2c), to represent the spatial variability of the temperature and relative humidity profiles.

$$\Delta x = x_{instrument-measured} - x_{model-simulated}, x = Radiance\ or\ BT, \quad (1)$$

$$\sigma_{\Delta x} = \sqrt{\sigma_{x_{instrument-measured}}^2 + \sigma_{x_{model-simulated}}^2}, x = Radiance\ or\ BT, \quad (2)$$

Randomly generated noise, accounting for the uncertainty in temperature and relative humidity, was added to the radiosonde
profiles for each case. In total, 1000 profiles were created with this random noise. The radiative Jacobians were used to
determine the radiance/BT difference between using the original radiosonde profiles and using the randomly generated profiles
as inputs. The standard deviation of the radiance/BT simulation from the generated 1000 profiles was utilized to represent the
1-sigma model-simulated uncertainty. In all uncertainty analyses in the following discussion, the sigma level is set to three
standard deviations (99.7% confidence level).

## 3 Results

### 3.1 AERI

The DLR observed by AERI is most strongly influenced by the near-surface atmospheric thermodynamic state. Quality control
of the AERI spectra was performed following Liu et al. (2022). For example, strong $CO_2$ and water vapor absorption channels
subject to calibration errors were excluded in this analysis following the Optical Depth Screening procedure of Liu et al.
(2022).

Figure 6 exhibits the AERI radiative closure test results. Overall, the uncertainty in the DLR bias for AERI mainly derives
from LBLRTM simulation uncertainties in the temperature-sensitive bands. In the window band, in FC2022 and FC2023,
measurement uncertainty dominates the total uncertainty, whereas, in FC2021, both measurement uncertainty and LBLRTM
simulation uncertainty contribute.

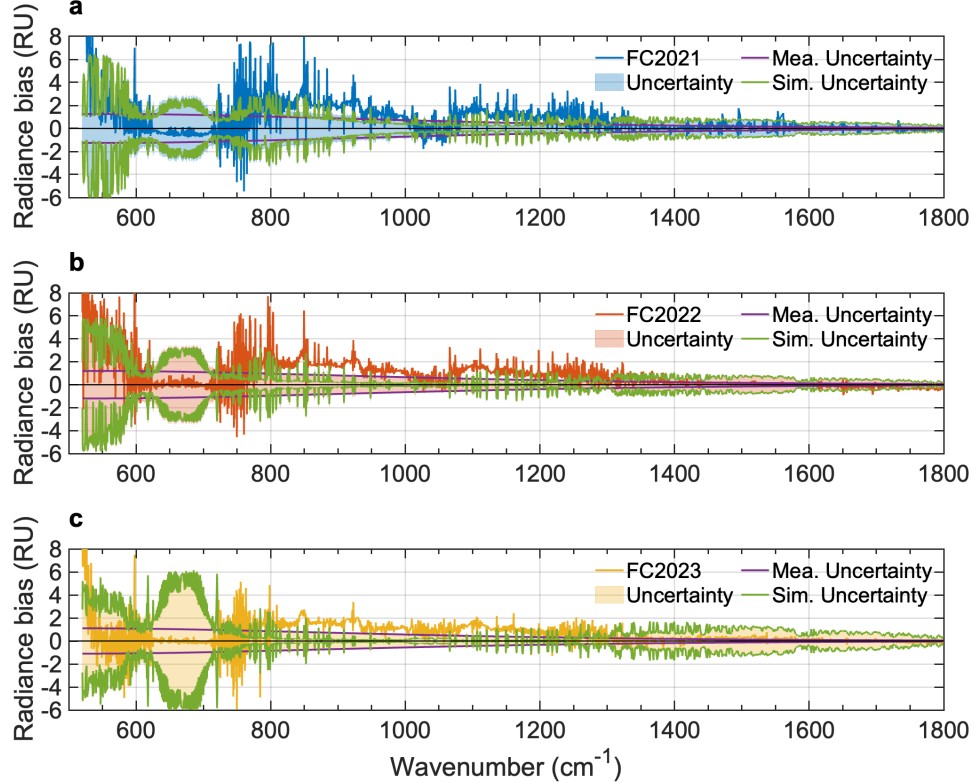

**Figure 6. AERI radiative closure test results. Each panel represents one field campaign. The blue line in panel a, the orange line in panel b, and the yellow line in panel c represent the DLR bias between 10-min averaged AERI-observed and LBLRTM-simulated spectra. The green lines and the purple lines represent the AERI measurement uncertainty and LBLRTM simulation uncertainty, respectively. The shadings represent the total DLR bias uncertainty.**

Good agreement between 10-min averaged AERI-observed spectra and LBLRTM-simulated spectra was observed in the $CO_2$ absorption band centered around 667 $cm^{-1}$ and the water vapor absorption band of 1400-1800 $cm^{-1}$, controlled primarily by atmospheric temperature, indicating excellent closure between the radiance measurements of AERI and the temperature profiles collected by radiosondes.

Over the three field campaigns, a persistent and stable positive DLR bias in the window band was detected, with the mean bias from the three campaigns (blue line in Figure 7) far exceeding their standard deviation (orange line in Figure 7). Moreover, the DLR bias in the window band in each of the field campaigns is larger than the DLR bias uncertainty (Figure 6). Because of the low BT in the window band, even a small radiance bias leads to a relatively large BT bias (Figure 7b). In this band, the radiance is primarily controlled by water vapor, aerosols, and clouds (Hansell et al., 2008; Seo et al., 2022). Through sensitivity tests (not shown), the bias was unlikely to be explainable by possible errors in the radiosonde water vapor measurements: over 150% of the original water vapor concentration in all vertical layers would be needed to remove this bias (not shown). The presence of optically thin aerosols or clouds of optical depth of ~0.06 at the altitude with relatively higher relative humidity





may explain the magnitude of this bias. However, the almost constant values of this bias across all three field campaigns make this hypothesis less likely.

It is interesting to note that historical AERI data measured elsewhere have also exhibited relatively large biases in the window band under clear-sky conditions (Liu et al., 2022). A FOV obstruction could introduce a positive radiance bias in the window band due to radiance leakage from the obstructive element having an emitting temperature higher than the scene temperature in the window band under clear-sky conditions (Turner, 2003). Based on a sensitivity test, the portion of obstructed FOV needed to explain this warm bias in the window band is around 2% (not shown). Since all three field campaigns targeted cold

and dry clear-sky atmospheric conditions whose calibration extrapolation process introduces larger uncertainties, it is also possible that calibration bias accounts for the radiance bias in the window band. Lower radiance in the window band draws the extrapolation further away from the blackbodies' emitted radiance, resulting in a larger calibration bias. However, whether the calibration process could lead to a consistent positive DLR bias in the window band is unknown.

As a result, a systematic, consistent warm radiance bias in the window band for AERI clear-sky observations is present, easily

removable for future retrieval analysis by subtracting the bias mean in channels whose radiance bias means (blue line in Figure 7a) are larger than their radiance bias standard deviation (orange line in Figure 7a); this is named the AERI warm bias correction.

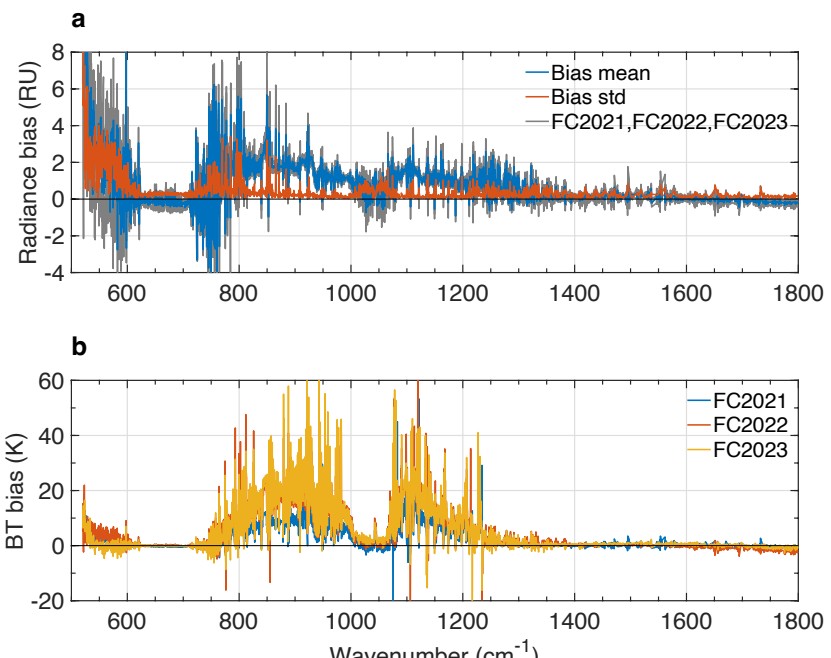

**Figure 7: AERI radiative closure test results. (a) DLR bias. The grey lines show the DLR difference between 10-min averaged AERI-**
**observed spectra and the LBLRTM-simulated synthetic spectra in the three campaigns. The blue line and the orange line represent the mean and standard deviation of the DLR differences, respectively. (b) BT bias.**





### 3.2 HiSRAMS

Radiative closure tests were performed on both the ground-based zenith-pointing measurements and the flight measurements of HiSRAMS. In light of the "roll-off" error in FC2021 measurements previously noted, the following discussions focus on

the results of FC2022 and FC2023, which show a better closure in both the oxygen and the water vapor absorption band at the frequency edges of each FFT spectrometer after the HiSRAMS refurbishment (Figure 8). The radiative closure results for ground measurements in FC2022 and FC2023 as well as flight measurements in FC2023 are shown in Figures 9 and 10, respectively. The two methods mentioned in Section 2.3 to determine the uncertainty of HiSRAMS ground measurements result in similar measurement uncertainties (purple lines in Figure 9), except for the significant measurement uncertainty at

the edge of FFT1 for both the oxygen and the water vapor band in FC2022, whose source is the remaining "roll-off" issue. This indicates that the frequency range with large measurement uncertainty, computed from the standard deviation of individual spectra, should be discarded in future retrieval analysis.

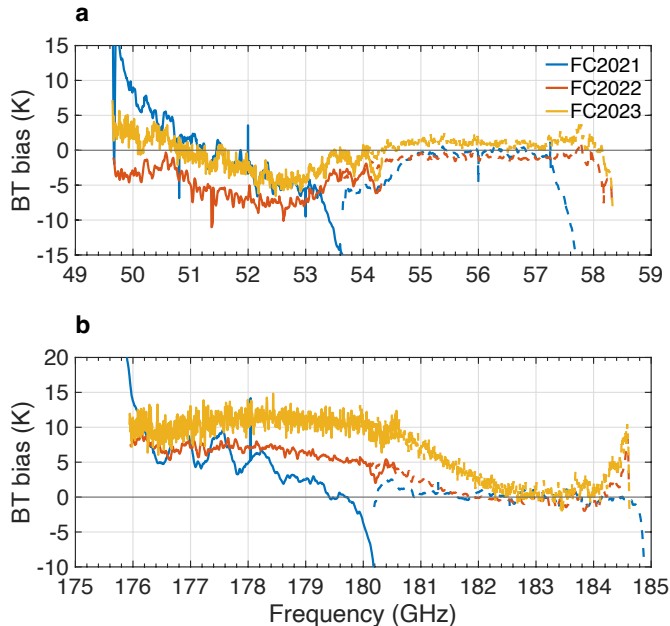

**Figure 8. HiSRAMS-observed ground-based zenith-pointing spectral brightness temperature bias for (a) oxygen band and (b) water**
**vapor band. Solid and dashed lines show the observed spectra from FFT0 and FFT1, respectively.**





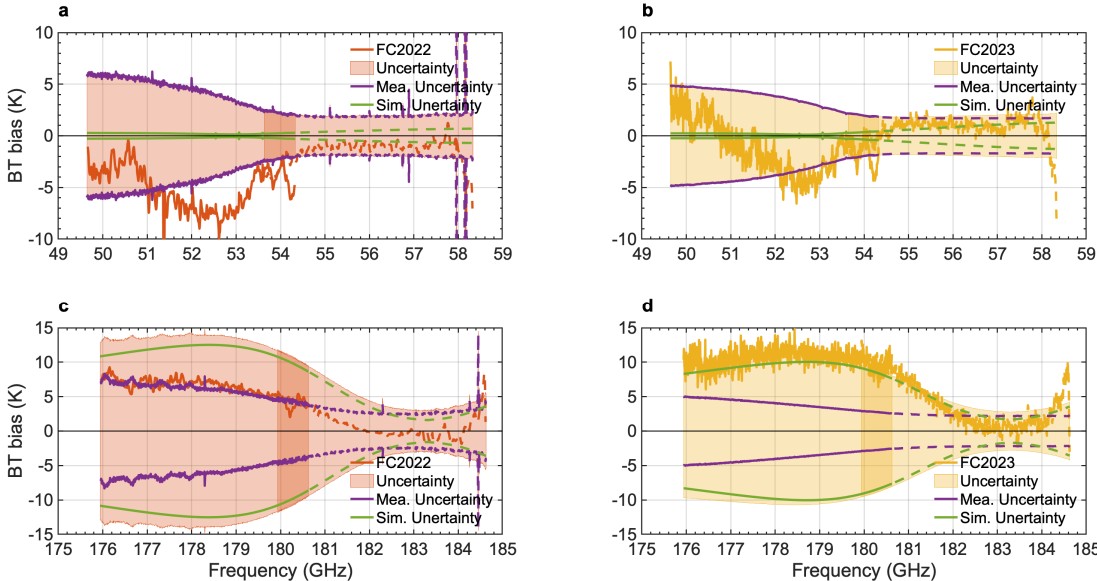

**Figure 9. The ground-based zenith-pointing HiSRAMS radiative closure test results for (a, b) oxygen band and (c, d) water vapor band. Orange lines in panels a and c and yellow lines in panels b and d represent the BT bias. In each panel, the shading represents the total uncertainty of the BT bias, while the purple and green lines represent the measurement uncertainty and simulation uncertainty respectively.**

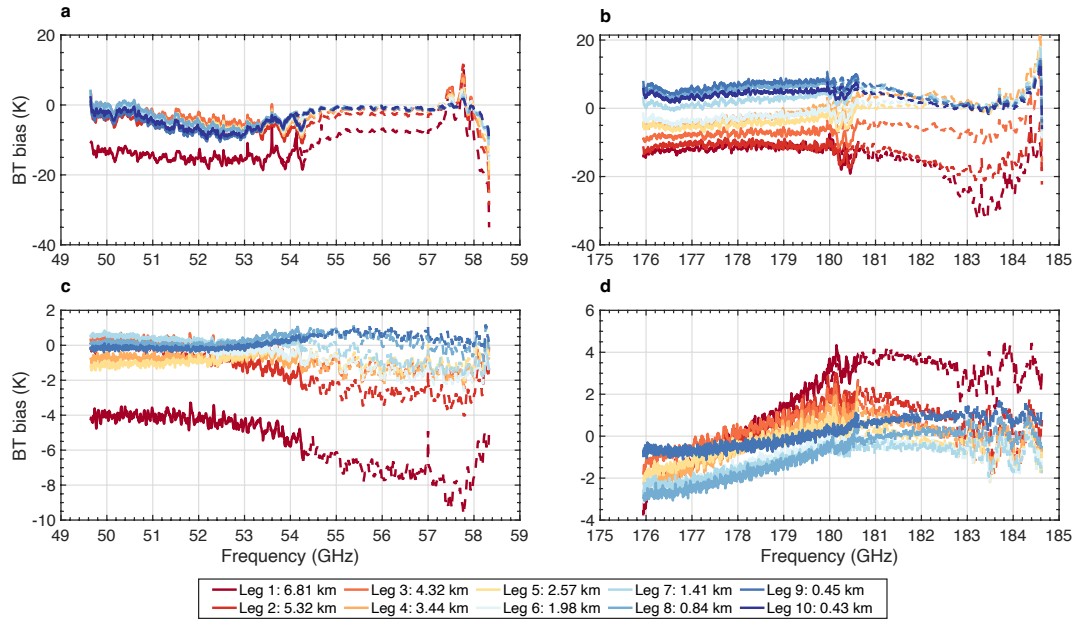

**Figure 10. BT bias for FC2023 flight measurements at different observational altitudes. (a, b) zenith-pointing BT bias in the oxygen and water vapor bands, respectively. (c, d) nadir-pointing BT bias in the oxygen and water vapor bands, respectively.**

The source of the radiative closure uncertainty in the zenith-pointing oxygen band radiometer, is attributed to the measurement

uncertainty. Uncertainties in the vertical temperature profiles are not significant in zenith-pointing HiSRAMS measurements. The zenith-pointing BT bias in the oxygen band (Figures 9a, 9b, and 10a), in the strong absorption frequency range (55-58 GHz) is relatively small: the BT bias in this frequency range is within the radiative closure uncertainty (Figure 9a, 9b). However, in the weak absorption channels (50-54 GHz), a significant BT bias occurs which exceeds the 3-sigma BT bias uncertainty. In FC2022 and FC2023, the BT bias for both ground and flight measurements has similar spectral shapes and

magnitudes (except for Leg 1 FC2023 flight measurements; these suffer from a large calibration bias, discussed later), suggesting a systematic bias, which may come from the calibration process. Considering all of the zenith-pointing BT biases in the oxygen band (except for Leg 1 FC2023 flight measurements), the mean BT bias is larger than the standard deviation of the BT biases (Figure 11), supporting the hypothesis that the bias may be systematic.

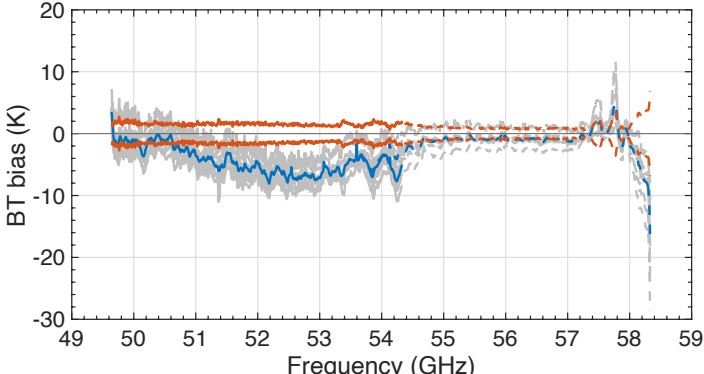

**Figure 11. HiSRAMS radiative closure results for the zenith-pointing oxygen band measurements from FC2022 and FC2023 ground measurements as well as FC2023 flight measurements. The grey lines represent individual BT biases for different conditions. The blue and orange lines represent the mean BT bias and the standard deviation of the BT biases, respectively.**

Compared to the oxygen band radiometer's zenith-pointing BT bias uncertainty, the measurement uncertainty as well as the simulation uncertainty contribute to the total uncertainty in the water vapor band radiometer's zenith-pointing BT bias. This

means that the zenith-pointing HiSRAMS measurements in the water vapor absorption band are sensitive to water vapor concentration. A relatively smaller BT bias was present in the strong water vapor absorption band (182-184 GHz) in zenith-pointing ground measurements (Figure 9c, 9d). There is a positive BT bias for both FC2022 and FC2023, with different magnitudes, in the weak absorption band at 176-180 GHz (Figure 9c, 9d). This bias is within the 3-sigma BT bias uncertainty. Measurements in different flight legs in FC2023 also show different BT biases in the water vapor absorption band (Figure

10b). Flight legs at lower altitudes tend to have positive BT biases; those at higher altitude legs tend to have negative BT biases, which suggests that these biases may be environment-dependent. The correlation coefficients between the environmental temperature from radiosonde temperature measurements and the channel-averaged BT biases for FFT0 and FFT1 in the water vapor band are 0.90 and 0.87, respectively (Figure 12), suggesting that the source of the HiSRAMS bias in the water vapor absorption band is related to the calibration processes.





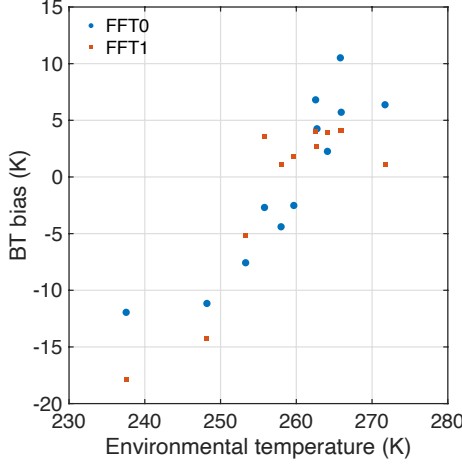

**Figure 12. Scatter plot between HiSRAMS zenith-pointing averaged BT biases in the water vapor band (FFT0 and FFT1) and environmental temperature from radiosonde measurements.**

A more accurate radiative closure was achieved for nadir-pointing HiSRAMS flight measurements (Figure 10c, d) compared to the zenith-pointing HiSRAMS flight measurements (Figure 10a, b). BT biases within 3 K were observed for nadir-pointing HiSRAMS measurements at all observational altitudes below 5.32 km.

Flight leg 1 (6.81 km) exhibits relatively poor radiative closure for all observational conditions and spectral ranges, which may be due to poor calibration accuracy in a cold environment. The HiSRAMS calibration process is sensitive to the environmental temperature; validation of the HiSRAMS calibration was performed in a well-controlled laboratory environment. However, the difference in environmental temperature during the flight measurements may introduce a larger bias to HiSRAMS measurements (Bliankinshtein et al., 2023a).

Due to the strong sensitivity of the zenith-pointing BT in the water vapor absorption band to vertical water vapor profiles, the uncertainty in the water vapor input results in the relatively large BT bias shown in Figures 9c, 9d, and 10b. This strong sensitivity could be beneficial to water vapor concentration retrieval if accuracy of the HiSRAMS zenith-pointing measurements under different environmental conditions can be assured; this requires more HiSRAMS ground-based and flight measurements.

### 3.3 Comparison of HiSRAMS and AERI radiative accuracy

As an established hyperspectrometer, AERI can be used to evaluate the accuracy of the HiSRAMS experimental radiometers. The BT biases in both AERI and HiSRAMS measurements are organized with respect to the total column optical depth for all the channels (Figure 13). In the original AERI measurements, the BT bias decreases overall with optical depth. The BT bias has a broader spread when the optical depth is low (Figure 13a); this may arise from the slight wavenumber mismatch between AERI observations and LBLRTM simulations. After the warm bias correction, a more accurate radiative closure of AERI is achieved (Figure 13b).





Nadir-pointing HiSRAMS measurements have a consistent radiometric characteristic across different optical depth ranges. The mean BT bias for nadir-pointing HiSRAMS measurements is relatively small, and the spread of the BT bias at different

optical depths is small (Figure 13c, 13d). On the contrary, the zenith-pointing HiSRAMS BT bias exhibits no simple relationship with optical depth. In the oxygen band, where the optical depth is relatively large, the BT bias is close to zero, showing good radiometric accuracy (Figure 13e). However, throughout other optical depth ranges in the oxygen band, and in the entire optical depth range in the water vapor band, the BT biases are large with a significant standard deviation.

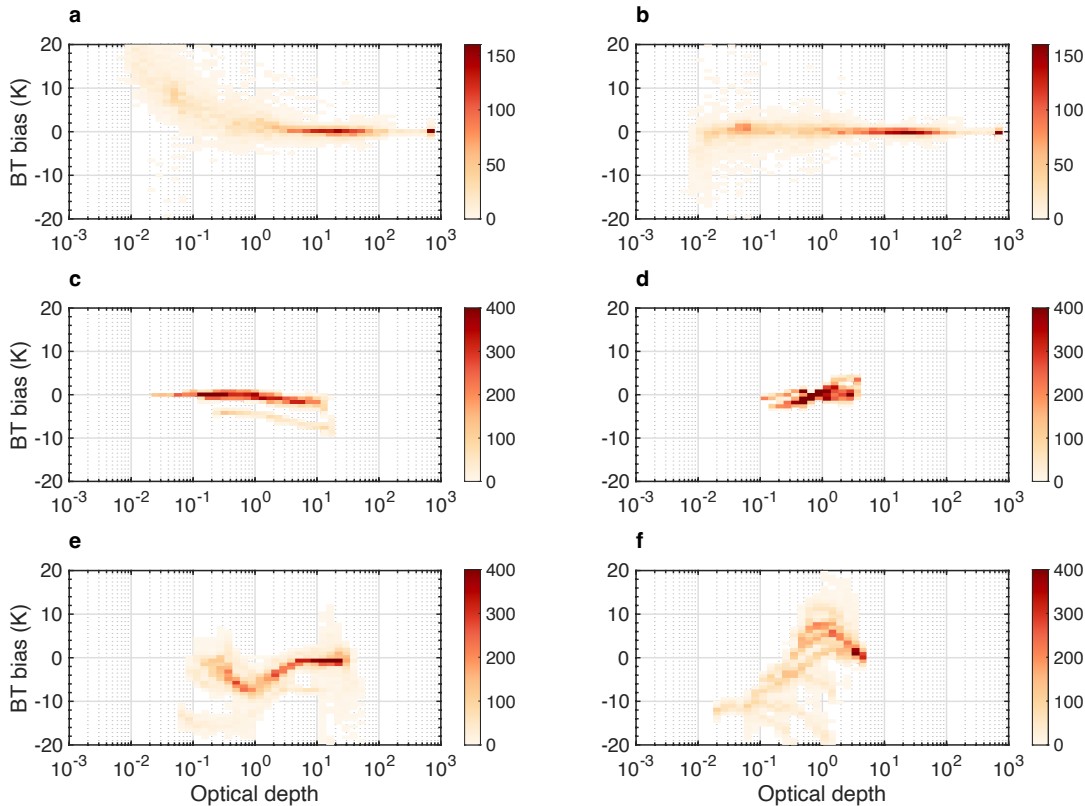

**Figure 13. BT biases with respect to optical depth at different channels for (a) AERI measurements, (b) corrected AERI measurements, (c) nadir-pointing HiSRAMS oxygen band measurements, (d) nadir-pointing HiSRAMS water vapor band measurements, (e) zenith-pointing HiSRAMS oxygen band measurements, and (f) zenith-pointing HiSRAMS water vapor measurements. The color represents the number of channels.**

Figure 14 compares the radiometric accuracy of AERI and HiSRAMS. The results for the mean BT bias and the standard

deviation of the BT biases at different optical depth ranges are shown. The optical depth here refers to the total column optical depth along the entire light path. Considering the corrected AERI radiometric accuracy as the benchmark, the nadir-pointing HiSRAMS measurements (yellow and purple dots and shadings in Figure 14) agree well with the corrected AERI measurements (orange dots and shading in Figure 14). The zenith-pointing HiSRAMS measurements (green and black dots





and shadings) clearly diverge from the corrected AERI measurements, indicating poorer radiometric accuracy. When
comparing the radiometric accuracy of AERI and HiSRAMS in zenith-pointing measurements, the viewing geometry of the
two instruments is identical, ensuring a fair comparison. However, when comparing the radiometric accuracy between AERI
zenith-pointing measurements and HiSRAMS nadir-pointing measurements, it is necessary to consider their different viewing
geometries, as this could also affect the radiometric accuracy.

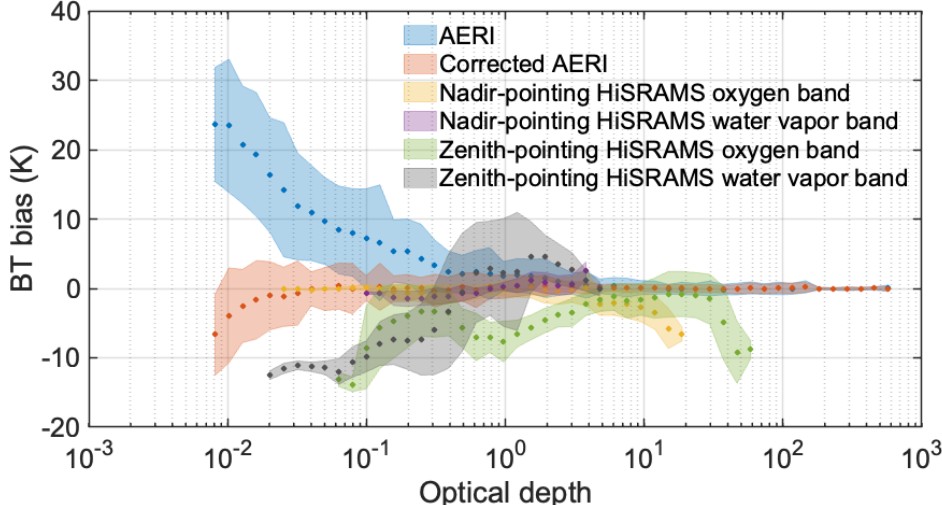

**Figure 14. Mean (dots) and standard deviation (shadings) of BT biases with respect to optical depth at different channels for AERI observations, corrected AERI observations, nadir-pointing HiSRAMS observations, and zenith-pointing HiSRAMS observations.**

In conclusion, nadir-pointing HiSRAMS measurements in the oxygen and water vapor bands have similar radiometric accuracy
to the AERI benchmark. However, poor radiometric accuracy has been observed in zenith-pointing HiSRAMS measurements
in oxygen and water vapor bands, indicating the necessity of improving HiSRAMS's zenith-pointing radiometric accuracy
calibration.

## 4 Conclusions and discussions

Vertical temperature and water vapor concentration profiles are essential for climate and weather studies. Hyperspectral
radiometers have been shown useful in retrieving high temporal and spatial resolution profiles of temperature and water vapor
concentration. Advancements in millimeter-wave technologies have made possible the development of hyperspectral
microwave radiometers exhibiting thousands of channels. HiSRAMS, designed and developed by an international team, is an
instance of such a development. The radiometric accuracy of this experimental instrument was evaluated in clear-sky
conditions, employing collocated clear-sky AERI and HiSRAMS spectral measurements, collected in Ottawa, Canada,
together with the radiosonde measurements of temperature and water vapor concentration profiles. Determining the



radiometric accuracy of the two HiSRAMS hyperspectral radiometers is a prerequisite for temperature and water vapor

concentration retrievals.

Three field campaigns were conducted to evaluate the radiometric accuracy of AERI and HiSRAMS. The radiance bias in the temperature-sensitive bands in AERI observations is relatively small, indicating a good accuracy of the temperature inputs from radiosonde measurements. A persistent warm bias in the window band was present in AERI measurements, which may be due to the FOV obstruction or calibration processes; this is easily corrected. Upon implementing the warm bias correction

in AERI measurements, a more accurate radiometric closure was achieved in the window band. HiSRAMS nadir-pointing spectra from flight measurements exhibit smaller BT bias compared to zenith-pointing spectra from both ground and flight measurements. Zenith-pointing HiSRAMS water vapor band measurements are sensitive to water vapor concentrations, illustrating the necessity of accurate HiSRAMS measurements for water vapor concentration retrievals.

A novel but straightforward method was developed to test the radiometric accuracy of the instruments based on the relationship

between radiative closure bias and total column optical depth. The radiometric accuracy of HiSRAMS was compared against the well tested instrument, AERI. Based on the BT bias at different optical depth ranges, nadir-pointing HiSRAMS measurements exhibit a radiometric accuracy comparable to AERI. However, poorer radiometric accuracy was observed in the zenith-pointing HiSRAMS measurements. To fully assess the source of this measurement bias, improved calibration and field campaigns are required.

The objective of designing and developing HiSRAMS is to test the retrieval performance of temperature and water vapor concentration from hyperspectral microwave observations in clear and cloudy sky conditions. This study focuses on the radiometric accuracy of HiSRAMS and AERI under clear-sky conditions as a first step. Future work includes a comparison of the temperature and water vapor retrieval performance between hyperspectral infrared and microwave radiometers under clear-sky conditions, assessing the synergy of HiSRAMS and AERI observations for temperature and water vapor retrieval under

clear-sky conditions, and validating the all-sky radiometric accuracy of HiSRAMS, as well as all-sky temperature, water vapor, and cloud retrievals based on HiSRAMS.

**Data availability**

The field campaign observational data together with the radiative forward model simulation data can be obtained from the Mendeley Data (https://doi.org/10.17632/kvt2s9ryk7.1).

**Author contribution**

YH conceived the research. YH, JG, and MW co-designed the measurement experiment. LL and YH developed the AERI forward model and performed AERI data collection and analysis. NB and PG developed HISRAMS forward model; NB and





SX performed HISRAMS data collection and analysis. LL led the writing of the manuscript with contributions from all co-authors.

**Competing interests**

The authors declare that they have no conflict of interest.

**Acknowledgments**

We acknowledge grants from the Canadian Space Agency (19FAMCGB16) and the Fonds de recherche Nature et Technologies (PR-283823) for the support of this work. We acknowledge the funding in support of HiSRAMS from the

European Space Agency (ESA contract 4000123417/NL/LA), and in support of AERI from the Canada Foundation for Innovation (CFI-36146), the Quebec Government, McGill University, and Université du Québec à Montréal. Many people from NRC, McGill University and Omnisys Instruments contributed to the successful completion of the measurement campaign. We thank the engineering, operation, and managerial staff from NRC and McGill University who made the project possible by working long hours during instrument deployments and field operations.

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
