# Peer review of "Radiative closure tests of collocated hyperspectral microwave and infrared radiometers"

_Atmospheric Measurement Techniques, 2023_

## Referee Comment (RC1)

**Radiative closure tests of collocated hyperspectral microwave and infrared radiometers, by Lei Liu et al.**

This is an interesting work which describes a radiative closure experiment in clear sky conditions performed by using collocated measurements collected with the HiSRAMS radiometers and the AERI Fourier transform spectroradiometer. Measurements were performed during three field campaigns: in 2021 and 2022 from ground, while in 2023 both from airborne and from ground. HiSRAMS is composed of two FTS radiometers operating in the oxygen band between 49 – 58 GHz and the water vapor band between 176 – 183 GHz; AERI is a well known Fourier spectroradiometer operating in the far and mid infrared portion of the Earth's emission spectrum, between 500 and 1800 cm$^{-1}$. The observations took place in zenith-pointing view with AERI from ground, and both in zenith- and nadir-pointing view with HiSRAMS. Simultaneous collocated radiosoundings  were launched during the three campaigns to measure the vertical profiles of temperature and water vapor. Simulations were performed by the authors to mimic the measurements of HiSMRAMS radiometers by using a validate code (Bliankinshtein et al. 2019) that uses the Rosenkranz gas absorption parameterization, and the well known code LBLRTM (Claugh et al. 2005) to mimic the AERI measurements. To estimate possible biases, the differences between the measurements and simulations were calculated with the associate standard deviations  by propagating, respectively, the spectral measurements uncertainties and the errors on the radiosondes, accounting also for the spatial inhomogeneity. The closure is well reached in the spectral bands below 800 cm$^{-1}$ and above 1200 cm$^{-1}$  simulating the AERI observations and  in the weak absorption bands of Oxygen (50-54 GHz) and Water Vapor (176-181 GHz), even though for water vapor the differences remains inside the 3-σ error. The bias with AERI comparison is attributed to the presence of very optically thin cirrus clouds or aerosols, the latter is attributed to the calibration procedure in zenith-pointing observations.

The paper is well structured and written, even though I suggest few corrections to enhance the clarity:

1)  Lines 92-94.  I would change the sentences in : "The small fluctuations in the temperature and water vapor profiles have a negligible effect in AERI and HiSRAMS detected radiances".

2) Lines 95-99. I would improve the discussion about the temperature inversions shown in the inset of Fig. 2a, if the authors want to mentioned it, it is fine but I think the description should be more accurate. For instance, from FC2021 (blu profiles)  to me seems there are more than two temperature inversion, at least one at around 2.5 km (T starts to increase again) stronger than the one at 1.2 km. Also, for FC2022 I can see at least two inversions, one at 0.5 km as pointed out, but also one at 2.5 km, similar to 2021, etc.. Maybe, if you could increase the grid vertical resolution on the y axes in both figures 2a and 2b would be helpful.

3) Line 119, "against" → "pointing"

4) I think it would be clearer indicating the actual radiance unit in all figures for AERI, I assume they are mW/(m2 sr cm-1), but it would be clearer if it would specified, in particular for me it is more helpful when I have to quantify the biases.

5) Section 2.2.1. Just a curiosity, I was wandering why you did not used the latest version 12.15 of LBLRTM, with the updated continuum 4.1 version?

6) Line 103, please write the coordinates in the standard form such as: [45.32° N, 75.66° W]

7) Line 112, "..200 level are inputs.." → "..200 levels are provided in inputs.."

8) Line 220, "errors" → "those" , "..inputs..", → ".. input profiles.."

9) Equations (1) and (2), I think it would be clearer if you could replace the x parameter with $R_v$ to indicate the vector of the radiances or brightness temperatures. Also, please shift the x → $R_v$ = Radiance of BT to the right side. Since the uncertainty of the measurements play a key role in the study and discussion of a radiative closure experiment, I suggest to explain in detail the components of the errors, for instance, from lines 220-222 I assume that the standard deviation on the model is obtained by summing in quadrature the 1-$\sigma$ error on the radiosondes profiles due to the instrumental error and that one due to spatial variability, is it so? In this is the case it would be really helpfull to follow if the authors could write the formulas for the errors propagation used.

10) I suggest to indicate the average biases arising for both instruments.

11)  Figure 9, I suggest to exchange figure 9c with 9b and viceversa to be coherent with the next figure 8. Also it would be really helpfull to indicate $O_2$ and WV band at the top of the two columns and zenit and nadir views horizontally.

12) Line 301, do not indicate Fig. 10a because I think is misleading here.

13) Line 313, ", the measurements uncertainty .." → ", both the contribution of the simulation and measurement uncertainty is not negligible.."

14) In Figure 12, please add the correlation coefficients.

References:

Bliankinshtein, N., Gabriel, P., Huang, Y., Wolde, M., Olvhammar, S., Emrich, A., Kores, M., and Midthassel, R.: Airborne Measurements of Polarized Hyperspectral Microwave Radiances to Increase the Accuracy of Temperature and Water Vapor Retrievals: an Information Content Analysis, AGU Fall Meeting Abstracts, A13K-2959, 2019

Clough, S. A., Shephard, M. W., Mlawer, E. J., Delamere, J. S., Iacono, M. J., Cady-Pereira, K., Boukabara, S., and Brown, P. D.: Atmospheric radiative transfer modeling: a summary of the AER codes, Journal of Quantitative Spectroscopy and Radiative Transfer, 91, 233-244, 10.1016/j.jqsrt.2004.05.058, 2005.

---

## Author Comment (AC1)

**The number of Lines in the comments corresponds to the original manuscript. The number of Lines in the responses (in red) corresponds to the revised manuscript with changes noted.**

Radiative closure tests of collocated hyperspectral microwave and infrared radiometers, by Lei Liu et al.

This is an interesting work which describes a radiative closure experiment in clear sky conditions performed by using collocated measurements collected with the HiSRAMS radiometers and the AERI Fourier transform spectroradiometer. Measurements were performed during three field campaigns: in 2021 and 2022 from ground, while in 2023 both from airborne and from ground. HiSRAMS is composed of two FTS radiometers operating in the oxygen band between 49 – 58 GHz and the water vapor band between 176 – 183 GHz; AERI is a well known Fourier spectroradiometer operating in the far and mid infrared portion of the Earth's emission spectrum, between 500 and 1800 cm-1. The observations took place in zenith-pointing view with AERI from ground, and both in zenith- and nadir-pointing view with HiSRAMS. Simultaneous collocated radiosoundings were launched during the three campaigns to measure the vertical profiles of temperature and water vapor. Simulations were performed by the authors to mimic the measurements of HiSMRAMS radiometers by using a validate code (Bliankinshtein et al. 2019) that uses the Rosenkranz gas absorption parameterization, and the well known code LBLRTM (Claugh et al. 2005) to mimic the AERI measurements. To estimate possible biases, the differences between the measurements and simulations were calculated with the associate standard deviations by propagating, respectively, the spectral measurements uncertainties and the errors on the radiosondes, accounting also for the spatial inhomogeneity. The closure is well reached in the spectral bands below 800 cm$^{-1}$ and above 1200 cm$^{-1}$ simulating the AERI observations and in the weak absorption bands of Oxygen (50-54 GHz) and Water Vapor (176-181 GHz), even though for water vapor the differences remains inside the 3-σ error. The bias with AERI comparison is attributed to the presence of very optically thin cirrus clouds or aerosols, the latter is attributed to the calibration procedure in zenith-pointing observations.

The paper is well structured and written, even though I suggest few corrections to enhance the clarity:

Thank you for the accurate summary of our work and your valuable comments, which have greatly contributed to improving our paper!

1) Lines 92-94. I would change the sentences in : "The small fluctuations in the temperature and water vapor profiles have a negligible effect in AERI and HiSRAMS detected radiances".
Done.

2) Lines 95-99. I would improve the discussion about the temperature inversions shown in the inset of Fig. 2a, if the authors want to mentioned it, it is fine but I think the description should be more accurate. For instance, from FC2021 (blue profiles) to me seems there are more than two temperature inversion, at least one at around 2.5 km (T starts to increase again) stronger than the one at 1.2 km. Also, for FC2022 I can see at least two inversions, one at 0.5 km as pointed out,

but also one at 2.5 km, similar to 2021, etc.. Maybe, if you could increase the grid vertical resolution on the y axes in both figures 2a and 2b would be helpful.

Thank you for your comment regarding the temperature inversion features observed in radiosonde measurements. This stands as a key feature distinguishing the clear-sky temperature retrieval performance between AERI and HiSRAMS, which will be described and discussed in our upcoming retrieval paper. Thus, we would like to retain this description within this manuscript. We have revised the discussion on temperature inversions, specifically in Lines 101 to 103. Additionally, we have updated Figures 2a and 2b with a higher vertical grid resolution.

3) Line 119, "against" → "pointing"

We revised the sentence in Lines 128-131.

4) I think it would be clearer indicating the actual radiance unit in all figures for AERI, I assume they are mW/(m2 sr cm-1), but it would be clearer if it would specified, in particular for me it is more helpful when I have to quantify the biases.

Thank you for pointing this out. One radiance unit (RU) equals to 1 mW/(m$^2$ sr cm$^{-1}$). We have specified the unit of AERI observed radiance in Line 128. For all the figures involved AERI observed radiance and/or radiance bias, we have included the unit in the captions.

5) Section 2.2.1. Just a curiosity, I was wandering why you did not used the latest version 12.15 of LBLRTM, with the updated continuum 4.1 version?

We began simulating the AERI-observed DLR in 2018. To maintain consistency with our prior work, we chose to employ version 12.9. Following your recommendations, we compared versions 12.9 and 12.16. The primary difference lies in the far-infrared spectrum. In the spectral range where significant radiance disparities exist between the two LBLRTM versions, the observed differences between simulations and actual observations are already large due to relatively inadequate calibration at the spectral detector's edge. Hence, we have retained the use of version 12.9 in this study. For ease of reference, Figure R1 has been included in the Supplement document (Section 1). We added this comparison in the manuscript in Lines 210-213.

[Figure]

Figure R1. (a) DLR difference between LBLRTM simulations using version 12.16 and version 12.9. (b) DLR difference between LBLRTM v12.16 simulations and AERI observations. (c) DLR difference between LBLRTM v12.9 simulations and AERI observations.

6) Line 103, please write the coordinates in the standard form such as: [45.32° N, 75.66° W]
We changed the description of the coordinates to Lines 70-71 in the standard form you suggested.

7) Line 112, "..200 level are inputs.." → "..200 levels are provided in inputs.."
Done.

8) Line 220, "errors" → "those" , "..inputs..", → ".. input profiles.."
Done.

9) Equations (1) and (2), I think it would be clearer if you could replace the x parameter with $R\nu$ to indicate the vector of the radiances or brightness temperatures. Also, please shift the $x \rightarrow R\nu =$ Radiance of BT to the right side. Since the uncertainty of the measurements play a key role in the study and discussion of a radiative closure experiment, I suggest to explain in detail the components of the errors, for instance, from lines 220-222 I assume that the standard deviation on the model is obtained by summing in quadrature the 1-σ error on the radiosondes profiles due to the instrumental error and that one due to spatial variability, is it so? In this is the case it would be really helpful to follow if the authors could write the formulas for the errors propagation used.

Thank you for the suggestions! We have revised Eq. 1 and Eq. 2 and now they are located in Lines 229 and 231, respectively. Yes, we combine the uncertainties in quadrature, which is similar to Eq. 2. To avoid the repetition, we included an additional sentence in Lines 241-242 to explain the method for combining these uncertainties.

10) I suggest to indicate the average biases arising for both instruments.
We have included the mean and standard deviation of BT biases for both instruments in Figure 13. The mean and standard deviation of HiSRAMS' BT biases cover the entire spectral range of each radiometer, whereas for AERI, they correspond to the channel associated with $CO_2$ and water vapor, respectively.

11) Figure 9, I suggest to exchange figure 9c with 9b and viceversa to be coherent with the next figure 8. Also it would be really helpful to indicate O2 and WV band at the top of the two columns and zenith and nadir views horizontally.
Thank you for the suggestions! We have updated Figure 9 to ensure consistency with Figures 5 and 10. Additionally, subtitles have been included in Figures 5 and 10.

12) Line 301, do not indicate Fig. 10a because I think is misleading here.
We updated the sentence in Line 330-332.

13) Line 313, ", the measurements uncertainty .." → ", both the contribution of the simulation and measurement uncertainty is not negligible.."
We updated the sentence in Lines 343-345.

14) In Figure 12, please add the correlation coefficients.
We have updated Figure 12.

References:
Bliankinshtein, N., Gabriel, P., Huang, Y., Wolde, M., Olvhammar, S., Emrich, A., Kores, M., and Midthassel, R.: Airborne Measurements of Polarized Hyperspectral Microwave Radiances to Increase the Accuracy of Temperature and Water Vapor Retrievals: an Information Content Analysis, AGU Fall Meeting Abstracts, A13K-2959, 2019
Clough, S. A., Shephard, M. W., Mlawer, E. J., Delamere, J. S., Iacono, M. J., Cady-Pereira, K., Boukabara, S., and Brown, P. D.: Atmospheric radiative transfer modeling: a summary of the AER codes, Journal of Quantitative Spectroscopy and Radiative Transfer, 91, 233-244, 10.1016/j.jqsrt.2004.05.058, 2005.

---

## Author Comment (AC2)

The number of Lines in the comments corresponds to the original manuscript. The number of Lines in the responses (in red) corresponds to the revised manuscript with changes noted.

This paper conducts a radiative closure exercise, using radiosondes observations to drive radiative transfer models which are then compared against both ground-based mid-infrared observations from the AERI and both ground-based and airborne observations from the microwave instrument HiSRAMS. Closure is considered achieved if the difference between the observation and radiative transfer calculation is within the uncertainties of both. Three different field campaigns were conducted, which had varying temperature and humidity conditions. The results demonstrated that there are times when closure was achieved, and other times when it was not. The authors then demonstrated how to compare the two instruments' closure using optical depth, arguing that the AERI and HiSRAMS agree when the HiSRAMS was airborne and looking downward, but that the did not agree when the HiSRAMS was on the surface looking upward.

Generally speaking, these closure tests are highly needed and should be conducted before thermodynamic profiles are retrieved from them (which the authors indicated is future work), so I commend them for this. However, there are some details that are missing and some concerns I have that should be addressed before this paper could be considered for publication.

Thank you for your comments and feedback to help us improve our paper! We have addressed and answered each of your concerns and comments point by point, following each respective paragraph.

One of my primary concerns is that this entire paper is focused upon three campaigns, where each one had a single radiosonde launch (Table 1). Thus, N=3. While careful quality control and processing of the radiosonde data can be done, the sampling uncertainty is still quite large.

Thank you for pointing this out! The sampling size is one of the key considerations in our radiative closure studies. We are consistently striving to conduct more field campaigns to gather additional data, as we consider this a crucial method for verifying the radiometric accuracy of hyperspectral instruments. These closure studies represent a fundamental step in advancing further research applications involving hyperspectral measurements.

During the three field campaigns we conducted, we observed persistent spectral features in the radiance differences between LBLRTM simulations and AERI observations. Considering the temporal gap between these campaigns, we believe that the spectral features are real. For these campaigns, the bias mean significantly exceeds the bias standard deviation illustrated in Figure 7a of the manuscript.

In Figure R1, we present the confidence level associated with each AERI channel. Across many channels, the sigma level exceeds 4, indicating a 99.9937% likelihood that the bias mean exceeds the bias standard deviation for these three field campaigns. We added the discussions in Lines 275-277.

[Figure]

Figure R1. The sigma level at each AERI channel in the window band (800-1200 cm$^{-1}$).

Another is that radiosondes, by their nature and how they are calibrated at the factory, have both systematic and random errors combined together when providing their uncertainty values (which the authors reported on line 86-88). Later, they tried to estimate the forward calculation uncertainty by Monte Carlo sampling the radiosonde data (line 225), attributing all of the uncertainties to random errors; it is not surprising to me that this did not result in much spread. I would highly recommend the authors consider apportioning the stated uncertainties into random and systematic errors as was done in Blumberg et al. JAMC 2017 (in the appendix), as correlated error will lead to larger impacts on the radiative transfer calculations.

Thank you for recommending this method! Following the instructions outlined in Blumberg et al. (2017), we have accounted for both the repeatability and reproducibility errors of the iMet-4 radiosondes utilized in this study:

repeatability errors (random errors):

$\sigma_{T,radiosonde\ repeatability} = 0.2K,\ \sigma_{RH,radiosonde\ repeatability} = 5\%$

reproducibility errors (systematic errors):

$\sigma_{T(P>100hPa),radiosonde\ repeatability} = 0.3K, \sigma_{T(P<100hPa),radiosonde\ repeatability} = 0.75K,$

$\sigma_{RH(T>0C),radiosonde\ repeatability} = 3\%, \sigma_{RH(-40C<T<0C),radiosonde\ repeatability} = 5\%.$

We have incorporated the spatial variability from ERA5 into the total random errors. An 8x8 grid box containing the trajectory of each balloon is considered to calculate the spatial variability, as suggested in the comment, to account for the smoothing impact in models. Thus:

$$\sigma_{T,random} = \sqrt{\sigma_{T,ERA5}^2 + \sigma_{T,radiosonde\ repeatability}^2}$$

$$\sigma_{RH,random} = \sqrt{\sigma_{RH,ERA5}^2 + \sigma_{RH,radiosonde\ repeatability}^2}$$

Following the procedures mentioned in Blumberg et al. (2017), the radiative closure results for AERI are shown in Figure R2. The radiance biases for all three field campaigns in the window band still exceed the total uncertainties at the 99.73% significance level.

According to the newly prescribed simulation uncertainty, we have identified a relatively broad uncertainty range in the temperature-sensitive channels ($CO_2$ absorption band centered at 667 $cm^{-1}$ and water vapor absorption band between 1400 and 1800 $cm^{-1}$). This implies that the radiative closure could be achieved in a wider range of the input variables.

We have updated the Figure 6 in the manuscript, the description of the input uncertainty in the manuscript (Lines 87-92, Lines 108-111), and the description of the AERI radiative closure results in Lines 262-264.

[Figure]

Figure R2: AERI radiative closure test results with updated simulation uncertainty. Here, the simulation uncertainty includes the random uncertainty from radiosonde measurements (repeatability errors) and enlarged spatial variability from ERA5 and the systematic uncertainty from radiosonde measurements (reproducibility errors).

The authors have identified an apparent bias in the infrared window channels of the AERI. I recommend they also read/reference the paper by Delamere et al. JGR 2010, which also noted a similar sized bias and did some exploratory work to understand it (albeit to no avail). One mechanism the authors should consider is that the MCT detector used in the AERI has a non-linearity correction applied, and perhaps this correction being applied is not correct.

Thank you for your comments regarding the warm bias detected in AERI observations within the window band! Previously, we referenced the paper by Delamere et al. (2010) solely in discussing the necessity of radiative closure tests (Line 64). We have included the reference of this paper when discussing the warm bias identified elsewhere in Line 286.

The AERI-122, i.e. the AERI utilized in this study, meets the nonlinearity requirement based on the third blackbody test cooled by liquid nitrogen, conducted as part of its certification testing at the University of Wisconsin Space Science Engineering Center in Madison, Wisconsin in November 2020. Additionally, we performed a third blackbody at ambient temperature calibration experiment and a FOV mapping experiment in August 2021, and AERI-122 successfully passed all tests. To further verify the stability of the instrument, future work of a third blackbody test cooled by liquid nitrogen is warranted to better address and constrain this issue. We specify the non-linearity-induced inaccuracy when addressing the persistent warm bias in the window band in Line 291.

More importantly, however, is that this apparent calibration error likely changes magnitude based upon the scene temperature; subtracting a bias offset is not appropriate for all scenes. A radiometric error is almost certainly associated with the slope relating detected signals to radiance (i.e., which is determined by the blackbody views), and as these are very cold scenes a small error in the slope will become larger as the sky temperature becomes colder (i.e., farther away from the ambient blackbody temperature). Thus, subtracting a bias only works for these cases, where the bias was determined directly from the radiosondes and the associated forward calculation, and is not something "easily removed" as suggested on line 270.

Thank for pointing out the subjective wording used in the manuscript. We aim to clarify that the magnitude of the radiance bias remains consistent across three distinct field campaigns. We have revised the manuscript to ensure objective language throughout.

Figure 10 and text around line 333: I am not convinced by the discussion associated with the bias seen in leg 1 on this flight. The environment at 6.8 km (leg 1) is colder than at 5.3 km (leg 2), but leg 3 at 4.3 km is also colder – yet the bias between legs 2 and leg 3 are very similar whereas the bias in leg 1 is a clear outlier. I believe something else is likely the problem, and that you should not try to simply state (as on line 333) that the issue "may be due to poor calibration in a cold environment."

We completely agree with your explanation regarding the cause of the bias in leg 1. We have revised the sentence in Lines 362-363. The reason we believe that the poor radiative closure for leg 1 could be attributed to poor calibration accuracy is that the brightness temperature observed at leg 1 is relatively lower compared to other legs (Figure 5c in the manuscript). This indicates that the scene temperature is relatively further from both the warm target temperature and ambient target temperature, potentially resulting in increased calibration error, such as non-linearity-induced inaccuracies. It emphasizes the necessity for further instrument improvement, which includes conducting a liquid nitrogen calibration experiment to identify the issue and potentially enhance the instrument's accuracy.

However, we are not sure whether this calibration issue alone could fully explain why leg 1 presents divergent radiative closure result. Considering the absolute outlier status of leg 1 in radiative closure, we would greatly appreciate any insights or possible explanations regarding the large bias observed in leg 1.

The comparison of the two instruments using optical depth is interesting. However, there are many trace gases that will be impacting the downwelling radiance observed by the AERI. Indeed, I can see in Figure 3 spectral structure associated with CFCs, which the authors indicated are not included in their downwelling calculations. I would highly recommend that the authors only use spectral channels from the AERI that are associated only with water vapor and carbon dioxide; spectral elements that have a contribution from other gases (CFCs, O3, CH4, N2O) should not be used in their optical depth plots. I suspect that it won't change the results too much, but makes the messaging much cleaner.

Thank you for the suggestion! We obtained the total optical depth at each AERI channels contributed by various greenhouse gases ($H_2O$, $CO_2$, $O_3$, $CH_4$, $N_2O$, and CFCs) based on FC2023 data. Each channel is labeled based on the variable contributing the most to the total column optical depth (Figure R3; see Section 2 in the Supplement document). This result has been included in the published netCDF file named "radiative_closure_tests_AERI.nc" (https://data.mendeley.com/datasets/kvt2s9ryk7).

Utilizing this channel selection index, we have updated Figures 13 and 14 by exclusively incorporating the channels labeled as water vapor and carbon dioxide for AERI measurements. Furthermore, in response to Reviewer 1's comment, we have also included the mean and standard deviation of BT biases in Figure 13. There are no significant differences between the old and updated versions of Figures 13 and 14.

[Figure]

Figure R3. AERI channel labels. Each AERI channel is labeled as the greenhouse gas who contributes the most to the total column optical depth.

Another large concern I have is associated with the surface contributions to the HiSRAMS observations when the radiometer is aloft and looking downward. You indicated that you are using the lowest flight level to effectively constrain the emission from the surface with observations; however, this was not discussed in any detail and thus we don't know if the effective skin temperature is realistic or not, and similar for the surface emissivity. This should be discussed in more detail.

We have provided detailed discussions on how we established the boundary conditions for HiSRAMS's nadir-pointing simulations in Lines 221-223. Microwave emissivity models require the input of a large number of surface parameters, such as surface type, soil moisture, vegetation characteristics, and surface roughness. Therefore, it requires an accurate surface property

database. To bypass that and to avoid the impact of the uncertainties in the surface emissivity, we decided to designate the nadir-pointing observations from the lowest flight level as the elevated surface, serving as the boundary constrain for other nadir-pointing observations.

This elevated surface emission already combines both the actual surface contribution and the atmospheric contribution between the real surface and this elevated surface. In other words, we consider the radiance observations from HiSRAMS at the lowest level as the "surface emission" at the elevated surface, already reflecting the effective skin temperature at this designated level.

Furthermore, you pointed out that the nadir-pointing HiSRAMS seems to agree well with the AERI but that the zenith pointing HiSRAMS does not. How much is the former results due to the way that the surface was constrained in the forward calculations? And in optically thin channels (e.g., 52-54 GHz), most of the observed radiance when airborne will be from the surface, and so are you really able to extract out the atmospheric optical depth well? What are uncertainties here?

The treatment of the surface contribution reduces the BT biases between nadir-pointing measurement and simulations, potentially leading to better radiative closure. At present, we are unable to quantify the exact impact of the boundary condition setting on radiative closure results for nadir-pointing measurements. Instead, we have included this discussion in the manuscript (Lines 387-389).

For HiSRAMS, the optical depth is derived from gas absorption coefficients parameterization for water vapor, oxygen, ozone, and nitrogen from Rosenkranz (2018, Rosenkranz, P.W.: Line-by-line microwave radiative transfer (non-scattering), Remote Sens. Code Library, doi:10.21982/M81013, 2017).

Minor details:

The authors need to mention the location of the intercomparison in the abstract and the paper.

Thank you for the reminder! We relocated the field campaign location details from Line 110 to Lines 70-71 in the manuscript. Additionally, we included the location description in the abstract (Line 15).

Line 102: using 9 adjacent gridboxes (i.e., a 3x3 grid around the desired point) in a model is not a good measure of the heterogeneity of a region, as virtually all models are diffusive and thus there will be a reasonable amount of correlation associated with model smoothing. Most models have significant model-induced correlations out to 5 times to 8 times the horizontal model grid spacing.

Thank you for pointing this out! We now use a 8x8 grid containing the trajectory of each balloon to calculate the spatial variability. The results of this analysis are presented in the previous discussion concerning the radiosonde measurement uncertainties. The new treatment of the spatial variability is includes in Lines 108-111.

Figure 6: it does not seem that the uncertainties from the observations and simulations are being combined properly in quadrature to get the total uncertainty. For example: Fig 6a at 850 cm-1 looks like the total uncertainty is the same as the measurement uncertainty; but as the simulated uncertainty is the same magnitude, the total uncertainty should be larger.

This is because the different elements in the figures are overlapping with each other. The enlarged figure (Figure R4) indeed shows that the total uncertainty in Figure 6a is larger than both the measurement uncertainty and the simulation uncertainty.

[Figure]

Figure R4. The AERI radiative closure results for FC2021 between 800 and 900 cm$^{-1}$.

Line 299: I think this phrasing for this sentence sounds better: "The primary contribution to the radiative closure uncertainty in the zenith…"

We have rephrased the sentence in Lines 326-327.

Line 315: this sentence seems very obvious.

We have deleted the sentence in Lines 345-346.

Fig 13c: please explain the second cloud of points here (which has a bias of -3 to -8 K)?

The second cluster of points in Figure 13c with a bias of -3 to -9 K corresponds to leg 1, where a relatively larger BT bias is observed.

References

Blumberg, W., Wagner, T., Turner, D., & Correia, J. (2017). Quantifying the accuracy and uncertainty of diurnal thermodynamic profiles and convection indices derived from the Atmospheric Emitted Radiance Interferometer. *Journal of Applied Meteorology and Climatology, 56*(10), 2747-2766.

Delamere, J., Clough, S., Payne, V., Mlawer, E., Turner, D., & Gamache, R. (2010). A far-infrared radiative closure study in the Arctic: Application to water vapor. *Journal of Geophysical Research: Atmospheres, 115*(D17).

---

## Referee Report (RR1)

**Radiative closure tests of collocated hyperspectral microwave and infrared radiometers,**

by Lei Liu et al.

The authors have addressed all the critical items I had pointed out, particularly I thank them for showing the comparison of the simulation with the new ones performed with the latest version of LBLRTM code v12.16.

From my point of view the manuscript can be published as is.

---

## Author Response (AR2)

The number of lines in the comments corresponds to the revised manuscript after the first-round revision. The number of lines in the responses (in red) corresponds to the revised manuscript with changes noted for the second-round revision.

**Reviewer 1:**

The authors have addressed all the critical items I had pointed out, particularly I thank them for showing the comparison of the simulation with the new ones performed with the latest version of LBLRTM code v12.16. From my point of view the manuscript can be published as is.

Thank you for your insightful comments towards our manuscript! Your suggestions have been invaluable in helping us refine and strengthen our paper.

**Reviewer 2:**

I thank the authors for their thoughtful consideration of my comments on the first draft of their paper. I find it much improved, and would recommend that it be published after the authors consider these few minor comments:

We are grateful for your valuable comments and suggestions regarding our manuscript. They have significantly contributed to the enhancement of our work.

1) Line 223: the surface contribution is also has a reflected term; i.e., surface_contribution = (downwelling radiance) * (1-emissivity) + (emission from surface). Using the lowest altitude observation as they are includes this, so only the text needs a slight tweak

Thank you for pointing this out! We have revised the manuscript in Lines 212-215.

2) Line 395: only AERI channels that dominated by either water vapor, carbon dioxide, or both are used in the analysis

Here, we have included only the channels that are dominated by either water vapor or carbon dioxide. We have revised the corresponding descriptions in the manuscript, specifically at Lines 357-359, Lines 377-378, and Line 392.

3) Line 425 (and this is absolutely critical to me): "Three field campaigns, during which a single radiosonde was launched, were conducted…". This is a good study, but its biggest weakness is the very small number of true comparisons that were done (3).

We acknowledge and agree with you that the limited number of field campaigns is a significant limitation of our study. Our belief in the authenticity of the phenomena described in the manuscript arises from the consistent radiative closure features observed across all three campaigns.

To address this limitation within the manuscript, we have incorporated the following two sentences at Lines 414-417:

"It is essential to acknowledge that the sampling size for this study was limited to three field campaigns. Despite the observed consistency in the radiative closure characteristics of AERI and HiSRAMS across these campaigns, further research involving additional field campaigns is necessary to comprehensively evaluate the radiative accuracy of these two instruments."

---

## Author Response (AR3)

**The number of lines in the comments corresponds to the revised manuscript after the second-round revision. The number of lines in the responses (in red) corresponds to the revised manuscript with changes noted for the third-round revision.**

Dear authors, thank you for revising the manuscript. Before I can accept the paper for publication, please fully address the last comment of the reviewer and mention in line 415ff that each field campaign consisted (only) of a single radiosonde. regards, Max Maahn

Thank you for your guidance and assistance during the peer-review process.

We have revised the sentences in Line 414-420: "It is essential to note that the sample size for this study was limited to three field campaigns, each accompanied by one radiosonde launch. The two instruments, HiSRAMS and AERI, are planned to be deployed in additional field campaigns and calibration experiments in the future, which will validate the closure assessment concluded here."